# Activated Inositol Phosphate, Substrate for Synthesis of Prostaglandylinositol Cyclic Phosphate (Cyclic PIP)—The Key for the Effectiveness of Inositol-Feeding

**DOI:** 10.3390/ijms25031362

**Published:** 2024-01-23

**Authors:** Antonios Gypakis, Stephan Adelt, Horst Lemoine, Günter Vogel, Heinrich K. Wasner

**Affiliations:** 1General Secretariat for Research and Innovation, GR-11527 Athens, Greece; agypa@gsrt.gr; 2Fachbereich C—Biochemie, Bergische University, 42119 Wuppertal, Germany; stephan.adelt@t-online.de; 3Institute for Laser-Medicine, Molecular Drug-Research Group, Heinrich-Heine-University, 40225 Düsseldorf, Germany; lemoine@uni-duesseldorf.de; 4BioReg Biopharm, Technology Innovation Laboratory, University of Illinois at Chicago, Chicago, IL 60612, USA

**Keywords:** cyclic AMP antagonist, cyclic PIP, *myo*-inositol, inositol phosphates, prostaglandylinositol cyclic phosphate, signal transduction, type 2 diabetes

## Abstract

The natural cyclic AMP antagonist, prostaglandylinositol cyclic phosphate (cyclic PIP), is biosynthesized from prostaglandin E (PGE) and activated inositol phosphate (n-Ins-P), which is synthesized by a particulate rat-liver-enzyme from GTP and a precursor named inositol phosphate (pr-Ins-P), whose 5-ring phosphodiester structure is essential for n-Ins-P synthesis. Aortic myocytes, preincubated with [^3^H] myo-inositol, synthesize after angiotensin II stimulation (30 s) [^3^H] pr-Ins-P (65% yield), which is converted to [^3^H] n-Ins-P and [^3^H] cyclic PIP. Acid-treated (1 min) [^3^H] pr-Ins-P co-elutes with inositol (1,4)-bisphosphate in high performance ion chromatography, indicating that pr-Ins-P is inositol (1:2-cyclic,4)-bisphosphate. Incubation of [^3^H]-GTP with unlabeled pr-Ins-P gave [^3^H]-guanosine-labeled n-Ins-P. Cyclic PIP synthase binds the inositol (1:2-cyclic)-phosphate part of n-Ins-P to PGE and releases the [^3^H]-labeled guanosine as [^3^H]-GDP. Thus, n-Ins-P is most likely guanosine diphospho-4-inositol (1:2-cyclic)-phosphate. Inositol feeding helps patients with metabolic conditions related to insulin resistance, but explanations for this finding are missing. Cyclic PIP appears to be the key for explaining the curative effect of inositol supplementation: (1) inositol is a molecular constituent of cyclic PIP; (2) cyclic PIP triggers many of insulin’s actions intracellularly; and (3) the synthesis of cyclic PIP is decreased in diabetes as shown in rodents.

## 1. Introduction

The number of inositol phosphates with regulatory properties ranges from inositol trisphosphate to inositol pyrophosphates [1,2,3,4,5], and you may ask yourself how many regulatory inositol phosphates need a living cell. Apart from this class of inositol phosphates, inositol is a component of cell surface glycans [6]. Bacteria synthesize di-inositol phosphates using CDP-inositol for the synthesis [7]. Living cells synthesize the cyclic polyol *myo*-inositol from glucose-6-phosphate, and additionally, inositol is adsorbed in the intestine from food [8]. Recent reports proclaim that inositol feeding has beneficial effects on the regulation of metabolism. *Myo*-inositol feeding improves illnesses such as polycystic ovary syndrome (PCOS), gestational diabetes, metabolic syndrome, and type 2 diabetes [9,10,11]. Inositol improves chronic inflammatory processes that can lead to cancer [12], and it plays a role in cellular energy metabolism [13]. Presently, it is largely unknown which biochemical pathways enable the curative effects of inositol feeding. A very likely explanation is that inositol feeding improves the impaired synthesis of the natural cyclic AMP antagonist prostaglandylinositol cyclic phosphate (cyclic PIP). The reasons are: (1) cyclic PIP contains myo-inositol as a molecular constituent; (2) cyclic PIP intracellularly triggers many of the anabolic regulations requested by insulin; and (3) diabetes is correlated with a decreased synthesis of cyclic PIP, as shown in rodents [14].

This report focuses on the group of inositol phosphates, containing a five-ring phosphodiester, though this group of compounds is presently nearly completely set aside. But cells do not synthesize these compounds without reason, and one may ask if this research field is not better cared for because of the high lability of these compounds and the resultant difficulty of working with them. In the early years of inositol phosphate research, scientists were engaged to find a physiological function for inositol (1:2-cyclic)-phosphate, which was then considered to be the dominant inositol phosphate, released from phosphatidylinositol on hormonal stimulation. When inositol 1,4,5-trisphosphate was shown to increase intracellular calcium levels, the research on inositol (1:2-cyclic)-phosphate ended. However, Philip Majerus and his group predominantly characterized the biosynthesis of inositol (1:2-cyclic,4)-bisphosphate and inositol (1:2-cyclic,4,5)-trisphosphate [15], and Sue Goo Rhee and his group studied the dependence of inositol cyclic phosphates, synthesized by phospholipase C-β, δ and γ (PLC-β, δ, and γ) from the experimental conditions and the presence of calcium ions [16]. Furthermore, when applying “mild experimental conditions”, the group of R. Michell could only isolate comparatively low amounts of inositol (1:2-cyclic,4,5)-trisphosphate, but much higher amounts of inositol 1,4,5-trisphosphate. This was taken as a strong point for the dominance of inositol 1,4,5-trisphosphate [17]. But at that time, it could not be considered that the detectable amount of inositol (1:2-cyclic,4,5)-trisphosphate could be low because it could be converted into a further, still unknown compound, apart from its fast hydrolysis into inositol 1,4,5-trisphosphate (see Section 3). This report shows that inositol (1:2-cyclic,4)-bisphosphate is the substrate for the synthesis of activated inositol phosphate (n-Ins-P), one of the two substrates for cyclic PIP biosynthesis. A prerequisite for the synthesis of n-Ins-P is that the inositol phosphate substrate contains a cyclic phosphodiester [14]. Last but not least, one has to be aware that inositol phosphates, containing a cyclic phosphodiester, are not degradation products of inositol phosphomonoesters. This argument, though it is widespread, does not obey the laws of thermodynamics. This means this ring-closure will not proceed spontaneously, because to form the five-ring phosphodiester, energy has to be added.

Earl Sutherland was convinced that living cells synthesize a second, intracellular regulator, which counteracts the regulations of cyclic AMP. This research led to cyclic PIP (Figure 1). The stimulation of its synthesis (a) by insulin or noradrenaline is shown in all rat organs and (b) by angiotensin II or vasopressin is shown in liver, heart, and kidney. Depending on the organ and the hormone, cyclic PIP synthesis increases several-fold within a few minutes and declines thereafter within 10 min to basal values [14,18]. Its chemical structure was identified through degradation experiments and mass spectrometry. Cyclic PIP is composed of prostaglandin E (PGE) and inositol (1:2-cyclic)-phosphate, which is bound by its C4-hydroxyl group to the C15-hydroxyl group of the PGE [14]. Its primary regulatory effects are seven-fold activation of protein ser/thr phosphatase and 100% inhibition of protein kinase A (PKA). Regulatory effects of cyclic PIP in living cells are for instance: 10-fold activation of glucose uptake into adipocytes; 90% inhibition of insulin release from pancreatic β-cells; termination of glucagon-stimulated autophagy and proteolysis in hepatocytes; and a 2.7-fold positive inotropic effect on the papillary muscle of the heart, which is connected with an elongation of the contraction time, whereas cyclic AMP triggers a shortening of the contraction time. In short, cyclic PIP switches off catabolism and switches on anabolism [14,19]. It is biosynthesized from prostaglandin E and activated inositol phosphate by cyclic PIP synthase, which is located in the light microsomal fraction [14]. Concerning the activated inositol phosphate, the following is discovered: (1) it is water-soluble and it elutes on gel filtration ahead of cyclic PIP, indicating that it has a molecular weight higher than 578 Dalton; (2) it contains no positive- but negative-charged groups; (3) it elutes from anion-exchange chromatography close to ATP, indicating that it most likely contains three phosphate groups, and the yield on this chromatography could not be increased above 20 to 40%, despite a maximally fast working-procedure; (4) it is significantly more labile than compounds containing a pyrophosphate structure; that means it is comparably labile as cyclic PIP, which decays by 80% within 30 min in a 0.5 molar salt solution; (5) differently to known water-soluble inositol phosphates, it adsorbs on charcoal, indicating that it contains a hydrophobic molecule group [20]. The essential question to be clarified is: How is n-Ins-P synthesized in living cells? Undoubtedly, it must be an activated inositol phosphate, since cyclic PIP synthase needs only prostaglandin E and activated inositol phosphate for the synthesis of cyclic PIP [21]. This report provides an initial answer to the biosynthesis of the activated inositol phosphate. Additionally, as a contribution to the special edition on the positive effects of myo-inositol supplementation, this report discusses why inositol supplementation can improve diabetes-related diseases.

## 2. Results

### 2.1. Prerequisites for the Biosynthesis of Activated Inositol Phosphate (n-Ins-P)

Rat liver was homogenized and separated into a particulate and a soluble fraction (10,000× *g*, 10 min). In one half of the two fractions obtained, the protein was denatured by boiling and the obtained sediment removed by centrifugation. The n-Ins-P content of the obtained four fractions was measured through its quantitative conversion to cyclic PIP [21]. The four single fractions, boiled and non-boiled supernatant and sediment, contained 15 to 35 pmol/assay n-Ins-P. No synthesis of n-Ins-P was found following incubation (15 min; 37 °C) of boiled sediment with non-boiled supernatant fraction. However, 175 pmol/assay n-Ins-P was synthesized upon incubation of the non-boiled particulate with the boiled, soluble fraction, indicating that the particulate fraction contains the enzyme, and the supernatant fraction contains the necessary substrates for n-Ins-P biosynthesis. N-Ins-P is an activated inositol phosphate [21], suggesting that a nucleoside triphosphate may be involved in its synthesis. To check this, the nucleotides of the soluble fraction were removed by treatment with charcoal. Subsequently, n-Ins-P is only synthesized by the particulate and the charcoal-treated supernatant fraction when GTP was added to the assay. This indicates that two substrates are needed for n-Ins-P biosynthesis, a nucleoside triphosphate and an inositol phosphate, which was tentatively named ‘precursor inositol phosphate’ (pr-Ins-P).

The best co-substrate was GTP, yielding 342 pmol n-Ins-P/assay. Setting this value at 100%, then the yield with UTP was 63%, with ATP 49% and CTP 37%. Various sediment fractions of rat liver were prepared (1100× *g*, 3600× *g*, 16,000× *g*, 107,000× *g*). The highest enzyme activity for n-Ins-P synthesis was found in the 3600× *g* fraction (61% of the summed-up activities of all fractions). N-Ins-P and pr-Ins-P both contain phosphates which may be sensitive to enzymatic dephosphorylation. The effect of the phosphatase inhibitor fluoride on the synthesis of n-Ins-P was determined. Fluoride increased the amount of n-Ins-P synthesized three-fold (Figure 2), but Li^+^, an inhibitor of inositol-mono-phosphatase, had no effect. The time dependence of n-Ins-P synthesis is shown in Figure 3. The amount of n-Ins-P increased for 2 min and then slowed as pr-Ins-P was depleted. Thereafter, it declined gradually by 50% despite the presence of fluoride indicating that n-Ins-P is degraded also by further processes.

### 2.2. Characterization of Precursor Inositol Phosphate (pr-Ins-P)

Pr-Ins-P could not be replaced by any of the commercially available inositol phosphates such as inositol (1,4)-, inositol (4,5)-, inositol (5,6)-, inositol (2,4)-bisphosphate, inositol (1,4,5)-, and inositol (1,3,5)-trisphosphate. Unlike these inositol phosphates, pr-Ins-P is as labile as n-Ins-P and cyclic PIP [20]. Increasing ionic strength caused rapid degradation (Table 1). It becomes completely degraded in 0.1 M HCl within 1 min. Pr-Ins-P is best kept at a neutral pH in a salt-free solution. A structural motif which shows such a lability is a five-ring phosphodiester structure [22]. To determine this, pr-Ins-P was incubated with snake venom phosphodiesterase under conditions applied for degradation of cyclic PIP [14]. This phosphodiesterase inactivates pr-Ins-P time-dependently (45% within 5 min), indicating that pr-Ins-P contains a phosphodiester structure, which can only be a five-ring phosphodiester due to its instability. Importantly, the enzyme, synthesizing n-Ins-P, accepts only inositol phosphate substrates which contain an inositol (1:2-cyclic)-phosphate structure.

In separate incubations, cyclic phosphodiesters were prepared of inositol (1,4)-, inositol (4,5)-, and inositol (5,6)-bisphosphate according to [23], which themselves are inactive to synthesize n-Ins-P. The obtained cyclized and non-cyclized inositol phosphate mixtures were not separated by chromatography, and the yields of the cyclized products were not determined. These product mixtures were substituted for pr-Ins-P in the assay for n-Ins-P synthesis. Only with inositol (1,4)-bisphosphate was a cyclic phosphodiester derivative obtained, which enabled the synthesis of n-Ins-P, indicating that pr-Ins-P could be inositol (1:2-cyclic,4)-bisphosphate. This compound was biochemically prepared [24] and the nearly pure compound replaced pr-Ins-P in the assay for n-InsP biosynthesis. The inositol (1:2-cyclic,4)-bisphosphate can be stored frozen without substantial decay for longer than one month. The shorter storage times observed with freshly prepared pr-Ins-P solutions most likely resulted from the presence of residual salts. Pr-Ins-P eluted in the range of inositol bisphosphates, ahead of cyclic PIP on Sephadex G15 chromatography. Treatment of pr-Ins-P-solutions with charcoal prior to the chromatography improved the yields, but there is no explanation for this finding. If no losses occurred during this purification, then the pr-Ins-P concentration in rat liver is about 0.5 µmolar.

### 2.3. Conversion of [^3^H] Inositol-Labeled pr-Ins-P to n-Ins-P and Cyclic PIP

Aortic myocytes [25], labeled with [^3^H] *myo*-inositol, were stimulated with angiotensin II for 0.5 min, which stimulates cyclic PIP synthesis and stimulates ahead the synthesis of the needed substrates [18]. The cells were then processed as described in the Section 4.3, and the pr-Ins-P containing extract was chromatographed on Sephadex G15. Two tritium-labeled substance peaks were resolved, the smaller one eluting in the range of inositol phosphates and the larger one eluting in the range of inositol (Figure 4A). Peak 1 contains [^3^H] pr-Ins-P in a yield of 65%, as determined by its conversion to [^3^H] n-Ins-P and then [^3^H] cyclic PIP.

Subsequent anion-exchange chromatography of peak I material resolved five [^3^H] inositol-containing substances (Figure 4B, curve with circle symbols). The material of peak I (fractions 23–27) was not identified (most likely it is inositol (1:2-cyclic)-phosphate). The very small peak II (fractions 32–35) is attributed to inositol mono-phosphates, the prominent peak III (fractions 41–46) to pr-Ins-P, peak IV (fractions 50–56) to inositol bisphosphates, and peak V (fractions 70–78) to inositol trisphosphates. As a control, the aortic myocytes were not stimulated with angiotensin II, and the subsequent anion-exchange chromatography (Figure 4B, curve with square symbols) revealed that angiotensin II stimulated the synthesis of pr-Ins-P about three-fold under the conditions applied.

High performance ion chromatography (HPIC) with an OH-based eluent system and suppressed conductivity detection is a sensitive and reproducible method for the analysis of inositol phosphates. Under the strong alkaline conditions applied, isomers with the same number of phosphate groups sometimes carry a similar net charge. This leads to difficulties in separating positional-isomers. An example is provided by the inositol bisphosphates, where up to now, only poor separations have been achieved [26]. It was found that the addition of 20% methanol as a modifier improves resolution. It is possible to resolve a mixture of inositol (1,6)- (peak 8), inositol (4,5)- (peak 9), and inositol (1,4)-bisphosphate (peak 10) on a solvent-compatible IonPac-HC column (Figure 5A). The assignment of these three inositol bisphosphates to the resolved peaks (8 to 10) was obtained through chromatography of the single compounds, and then of various mixtures of these three compounds. Acid hydrolyzed, [^3^H] inositol-labeled pr-Ins-P was applied to this chromatography. It elutes between 19 and 19.5 min, in the range of inositol (1,4)-bisphosphate (Figure 5B), which peaks after 19.3 min. This confirms that pr-Ins-P in all likelihood is inositol (1:2-cyclic,4)-bisphosphate.

### 2.4. Preparation of [^3^H] Guanosine-Labeled n-Ins-P and Its Use in Cyclic PIP Synthesis

The synthesis of n-Ins-P was carried out with [^3^H] GTP and unlabeled pr-Ins-P. The product was diluted with unlabeled n-Ins-P and chromatographed on Sephadex G15 to separate the substrate [^3^H] GTP (MW 523), and its degradation products [^3^H] GDP (MW 443) and [^3^H] GMP (363) from the synthesis product, [^3^H] guanosine-labeled n-Ins-P (most likely MW 667), which elutes ahead of GTP. The re-chromatography shows that [^3^H] guanosine-labeled n-Ins-P (Figure 6A) was obtained. The little shoulder eluting subsequent to n-Ins-P most likely is [^3^H] GTP, which was not completely separated on the first gel filtration. The control experiment was performed in the same way, but in the absence of pr-Ins-P. In this case, no [^3^H] guanosine-labeled n-Ins-P could be synthesized, and no [^3^H]-labeled material was found in the n-Ins-P containing fractions. The [^3^H]-labeled guanosine material peaking in fraction 38 is most likely unconsumed [^3^H] GTP (Figure 6B, curve with squares).

Cyclic PIP synthase assays were performed as described [20,21], using three different radioactively labeled substrates. This means tritium-labeled was either the PGE or the activated inositol phosphate, which contained either a tritium-labeled inositol or guanosine molecule-part. The first assay contained [^3^H] PGE_1_ and unlabeled n-Ins-P as substrates. In the subsequent silicic acid chromatography (Figure 7A), unreacted [^3^H] PGE was eluted in fraction one and two, and the product [^3^H] cyclic PIP in fraction four and five [20]. The second assay contained unlabeled PGE and [^3^H] guanosine-labeled n-Ins-P as substrates. In the silicic acid chromatography (Figure 7B), no [^3^H] guanosine-labeled compound was eluted in the cyclic PIP-containing fractions four and five, but [^3^H] guanosine-labeled material was eluted predominantly in fraction six with water, and was identified as [^3^H] GDP by co-chromatography with [^14^C] GDP by anion-exchange chromatography (inset of Figure 7B). The third assay contained unlabeled PGE and [^3^H] inositol-labeled n-Ins-P as substrates. In silicic acid chromatography (Figure 7C), all [^3^H] inositol-labeled material was eluted in fraction four and five. In all three assays, the amount of n-Ins-P supplied limited the reaction, and the synthesis of cyclic PIP synthesis stopped as n-Ins-P was consumed. In case of the second assay, the possibility that the identified [^3^H] GDP could have been present as an impurity from synthesis of the [^3^H] guanosine-labeled n-Ins-P is dispelled, because the synthesized [^3^H] guanosine-labeled n-Ins-P is separated from [^3^H] GTP and all the more from GDP, which elutes six fractions later than GTP on Sephadex G15 chromatography. Additionally, in order to be sure that all nucleotides are separated, only the front 50% of the [^3^H]-guanosine-labeled n-Ins-P peak (Figure 6A) was collected for further assays. The obvious interpretation is that cyclic PIP is synthesized by condensation of the inositol (1:2-cyclic)-phosphate part of n-Ins-P with PGE, and the GDP-part of n-Ins-P is simultaneously split off.

## 3. Discussion

An essential step in elucidating the biosynthesis of cyclic PIP (Figure 1) was to find after prostaglandin E—as the first substrate — the second substrate, which was initially named novel, and then activated inositol phosphate n-Ins-P [20,21]. For the biosynthesis of n-Ins-P are needed a nucleoside triphosphate and the precursor inositol phosphate, which is chemically as labile as n-Ins-P and cyclic PIP. This lability is attributed to the five-ring phosphodiester. It remains, presently, unsolved whether the decline of n-Ins-P synthesis (a) at longer incubation times of the time-dependence of cyclic PIP synthesis (Figure 3) and (b) at higher fluoride concentrations (Figure 2), results from hydrolysis of the five-ring phosphodiester under the experimental conditions. From [^3^H] inositol-labeled aortic myocytes, [^3^H] inositol-labeled pr-Ins-P was isolated in a yield of 65% despite its instability (Figure 4). This high yield of [^3^H] inositol-labeled pr-Ins-P indicates that the synthesis of pr-Ins-P, n-Ins-P, and cyclic PIP represents a dominant path of inositol metabolism, at least under the applied experimental conditions of hormonal stimulation of cyclic PIP synthesis. There is currently no explanation why Majerus and Rhee, experimenting with primarily purified PLCs, obtained relatively low yields of inositol cyclic phosphates [15,16] when compared to the yields obtained with living cells that were stimulated with hormones (see Figure 4). The difference is that in the cells the PLC is activated by hormones [20].

Pr-Ins-P is in all likelihood inositol (1:2-cyclic,4)-bisphosphate. The points for this conclusion are: (1) acid hydrolyzed pr-Ins-P co-chromatographs on HPIC with inositol (1,4)-bisphosphate (Figure 5B); (2) chemically and especially biochemically synthesized inositol (1:2-cyclic,4)-bisphosphate substitute pr-Ins-P; and (3) periodate oxidation of the phosphomonoester of cyclic PIP causes complete breakage of the molecule [20]. This cleavage can only occur when the substituents PGE and phosphate are bound to the inositol in trans-position. These are obviously the C1 and C4 positions, since inositol (1:2-cyclic)-phosphodiester hydrolyses by 85% to inositol 1-phosphate.

Inositol phosphates with a cyclic phosphodiester structure have attracted only minor attention so far. Majerus suggested that the inositol phosphomonoesters serve as regulators of intracellular Ca^2+^ levels, while inositol phosphates containing cyclic phosphodiester structures may fulfill other cellular functions [15]. The existence and action of cyclic PIP supports this suggestion. On hormonal stimulation of living cells, a confusing amount of different inositol phosphates appears to be intracellularly formed or correctly said is isolated. In order to compare different reports, it would be advantageous if all experiments on isolation of inositol phosphates would apply for instance the “mild procedure” used by R. Michell [17]. The reason is that the isolation procedure very likely has an impact on the obtained results, because for instance, inositol cyclic phosphates are hydrolyzed quickly, and phosphates can rearrange their position on the inositol at an acidic pH value. Furthermore, there appears to be no selectivity, which means one would expect release of primarily one inositol phosphate that is either itself regulatory active or that is converted into an active product in a further reaction. With respect to the multitude of inositol phosphates isolated, one may ask for a mechanism that regulates which inositol phosphate is predominantly synthesized at a time. Presently, such a mechanism is not found or non-existent. Briefly said, there are inositol phosphates, which are end products like inositol 1,4,5-trisphosphate. Then, there are inositol phosphates, which are substrates for the synthesis of other compounds, such as pr-Ins-P, inositol (1:2-cyclic,4)-bisphosphate, for the synthesis of n-Ins-P, and cyclic PIP. Then, there are inositol phosphates, which are most likely degradation products from primary formed inositol phosphates, such as inositol 1,4-bisphosphate, the hydrolysis product of pr-Ins-P and inositol 1-phosphate, a degradation product of n-Ins-P, and cyclic PIP. Last but not least, it is tempting to ask if the homolog of inositol (1:2-cyclic,4)-bisphosphate, inositol (1:2-cyclic,4,5)-trisphosphate, could be the substrate for synthesis of a further regulatory compound.

Inositol diphosphate (pyrophosphate) mixtures obtained from inositol (1,4)- and inositol (2,4)-bisphosphate, which themselves are not recognized by cyclic PIP synthase as substrates, can be substituted for n-Ins-P to synthesize cyclic PIP analogues. They react at a slower rate than n-Ins-P [20]. Additionally, cyclic PIP is synthesized from PGE and n-Ins-P without the need for ATP [21]. These results indicate that n-Ins-P must be an inositol phosphate derivative that contains the energy to form the allyl-ether bond of cyclic PIP, and that this activation is most likely associated with a diphosphate motif. The findings (1) that nucleoside triphosphates are necessary for n-Ins-P biosynthesis; (2) that [^3^H] guanosine is incorporated into n-Ins-P (Figure 6A), and (3) that [^3^H] GDP is released from [^3^H] guanosine-labeled n-Ins-P in the course of cyclic PIP synthesis (Figure 7B), indicate that n-Ins-P in all likelihood is guanosine diphospho-4′-inositol (1′:2′-cyclic)-phosphate (Figure 8). The result of these experiments looks plausible; nevertheless, more experiments are necessary to make this result indisputable. The final proof of this proposed chemical structure of n-Ins-P is the chemical synthesis of n-Ins-P, which is still in the planning stage.

In summary, pr-Ins-P, inositol (1:2-cyclic,4)-bisphosphate and GTP, with the elimination of pyrophosphate, are combined to activated inositol phosphate, guanosine diphospho-4-inositol (1:2-cyclic)-phosphate, whose inositol (1:2-cyclic)-phosphate part and PGE are combined by cyclic PIP synthase to cyclic PIP, releasing the GDP-part of n-Ins-P. This reaction sequence is comparable with the synthesis of UDP-glucose from glucose 1-phosphate and UTP, the transfer of glucose to any glucose-acceptor, and the release of UDP. The difference is that in this case, a glycosidic linkage is formed, whereas in the biosynthesis of cyclic PIP, a certainly not-very-common allyl-ether bond is formed.

Yeast cells, deficient in *myo*-inositol synthesis, are dependent on external support with inositol in order to stay alive. This is taken as an indication that *myo*-inositol and its derivatives are essential for the viability of cells [27]. Inositol-feeding has been shown to be an effective cure — for instance in the case of diabetes-related illnesses such as PCOS and gestational diabetes [28,29]. Currently, not much is known about which defects can be bypassed by inositol feeding. Furthermore, the decreasing synthesis of cyclic PIP is the most likely reason that initiates insulin resistance [30]. But the existence and action of cyclic PIP has not really been acknowledged yet. Nevertheless, cyclic PIP is most likely the key to understanding why inositol feeding has positive effects on illnesses related to insulin action. It can be assumed that inositol-deficiency will lead to decreased synthesis of phosphatidylinositol and phosphoinositides, and consequently of activated inositol phosphate, and last of cyclic PIP. However, the synthesis path from *myo*-inositol to phosphatidylinositol, and moreover to activated inositol phosphate and cyclic PIP, cannot be defective, because then, inositol-feeding could not improve the decreased cyclic PIP synthesis, which is observed in diabetic rodents [14], and which will most likely also be found in diabetic humans. It follows that the defect, causing inositol deficiency, must lie ahead of this reaction path. This means that inositol deficiency most likely results from its malabsorption by the intestine, and also from its increased excretion by the kidneys [31,32]. These defects of absorption and excretion of inositol are suggested to result from increased glucose levels [33]. Then, inositol deficiency most likely develops as a result of elevated blood glucose levels in advanced type 2 diabetes. Thus, inositol-feeding will prevent the continuous deterioration of the disease, but it will most likely not cure it. The defect causing insulin resistance lies most likely primarily between the insulin receptor and cyclic PIP synthase, whose activity is dependent on tyrosine-phosphorylation and de-phosphorylation and serine/threonine phosphorylation and de-phosphorylation, as discussed in [19]. Nevertheless, inositol feeding will be helpful as long as no medications are available, which optimally help type 2 diabetic patients. That means medications which prevent that these patients develop all the life-threatening comorbidities. Such a medication will not only lower elevated blood glucose levels, but will also switch on anabolism like insulin does. Thus, type 2 diabetic patients will be better treated with a combination of inositol feeding and treatment with drugs like metformin, since inositol feeding will help to provide one of the substrates for cyclic PIP synthesis and metformin will activate cyclic PIP synthase [19]. There are already reports which found that this combined treatment is helpful [34].

## 4. Materials and Methods

### 4.1. Materials

Prostaglandin E_1_ was obtained from Cayman Chem. Comp. (Ann Arbor, MI, USA); BCA-protein-assay-reagent from Pierce (Rockfield, IL, USA); pentobarbital-sodium (Nembutal R) from Wirtschaftsgenossenschaft dtsch. Tierärzte (Hannover, Germany); ATP, GTP, UTP, CTP, ADP, and GDP were obtained from Roche Diagnostics (Mannheim, Germany); inositol(1,4)P_2_, inositol(2,4)P_2_, inositol(4,5)P_2_, inositol(5,6)P_2_, inositol(1,4,5)P_3_, noradrenaline, adrenaline, and silicic acid SIL-350 were purchased from Millipore-Sigma (Darmstadt, Germany); Sephadex G15, HL-Q-Sepharose HP, [5,6(n)-^3^H] PGE_1_, [8-^3^H] GTP, [8^-14^C] ADP, [2-^3^H] inositol with stabilizer PT6-271, [2-^3^H] inositol(1,4)P_2_, and [^3^H] inositol(1,4,5)P_3_ were obtained from GE-Healthcare (Solingen, Germany); activated charcoal, toluene, ethyl acetate, methanol, Rotiscint 11, and Rotiscint 22 were obtained from Carl Roth (Karlsruhe, Germany); all other chemicals of reagent grade were from E. Merck (Darmstadt, Germany).

A regiospecific ionsitol-P_5_/inositol-P_4_-phosphohydrolase from *Dictyostelium discoideum* was exploited for the synthesis of inositol (1,6)-bisphosphate [35].

Figure 2, Figure 3, Figure 4, Figure 5, Figure 6, Figure 7 and Figure 8 were drawn with Microsoft office Excel 2007 (12.0.6787.5000) SP3 MSO (12.0.6785.5000).

### 4.2. Animals

Male Sprague–Dawley rats (200–300 g) had free access to water and food (standard diet Ssniff (Soest, Germany)). They were anesthetized with an overdose of sodium pentobarbital (30–40 mg/kg body weight) prior to use.

### 4.3. Preparation of Precursor Inositol Phosphate (pr-Ins-P)

Rat liver (11–13 g), extra-corporally perfused with oxygenated (95% O_2_, 5% CO_2_), physiological saline (0.9%, *w*/*v*), was stimulated with 5 × 10^−5^ M noradrenaline for 50 s and then perfused for 10 s with water in order to wash out the salts. Thereafter, the liver was homogenized in 25–30 mL of ice-cold water with a Dounce and the homogenate poured into a mixture of chloroform/methanol (2:1, *v*/*v*) present in twofold excess. The phases were separated by centrifugation (10,000× *g* for 10 min at 4 °C). The water phase, after removal of remaining organic solvents by rotation evaporation for 5–10 min and adjustment of the pH to 6.5–7, was treated with acid- and ethanol-washed activated charcoal (3 g charcoal in water of pH 7/liver). After removal of the charcoal by centrifugation (15,000× *g* for 10 min at 4 °C), the UV-absorbance of the solution was determined at 254 nm. In case of detectable UV-absorbance, the solution was treated a second time with charcoal. (For isolation of [^3^H] inositol-labeled pr-Ins-P from aortic myocytes (2–3 × 10^7^ cells), 60–100 mg charcoal was used.) Thereafter, the solution was lyophilized.

### 4.4. Gel Filtration and Anion-Exchange Chromatography of pr-Ins-P

The chromatography was performed on Sephadex G15 (2.6 cm × 100 cm column) with a flow rate of 6–8 mL/min at 8 °C and double distilled water as eluent because of the lability of the 5-ring phosphodiester. The conductivity as a measure of ionic molecules and the UV absorbance of the eluent at 254 nm was determined. (On gel filtration of charged low molecular substances their hydration volume plays an essential role. For instance, chloride and bromide ions have the same negative charge, but the smaller chloride ion attracts more water than the bromide and elutes more than 10 fractions ahead of the bromide ion from the column. Furthermore, the elution depends on the concentration of the chromatographing solution. This means that substances applied to the column in a high concentrated solution elute later than substances of a low concentrated solution. Most likely, the hydration volume decreases with the increasing concentration.) In a control chromatography, inositol trisphosphate peaks in fraction 38, n-Ins-P in fraction 54, ATP in fraction 56, and cyclic PIP in fraction 60.

Anion-exchange chromatography at 8 °C was performed on a HL-Q-Sepharose HP (1.5 cm × 25 cm column), which was equilibrated with 10 mM acetate buffer (pH 5.5) and eluted with a linear gradient from 0 to 250 mM KBr (2 mL/min flow rate) in a total volume of 400 mL. The desalting of the pr-Ins-P containing solution was performed by gel filtration, since the bromide ions eluted more than 10 fractions after pr-Ins-P from the column [14]. In a control chromatography, AMP peaked in fraction 31, ADP in fraction 49, and ATP in fraction 61.

### 4.5. Assay of pr-Ins-P and n-Ins-P

The assay contained 30 mM Tris/HCl buffer (pH 7.5), 100 mM fluoride, 4 mM GTP, 10 mM Mg^2+^, 1.8 mM mercaptoethanol, 50–200 µL solution of pr-Ins-P and 50 µL particular enzyme (0.1–0.2 mg protein) in 0.5 mL volume. The reaction was stopped after 2 min by addition of 1 mL chloroform/methanol (2:1, *v*/*v*); the formed precipitate was sedimented (2000× *g* for 10 min at 20 °C); the water phase was removed and lyophilized. The residue obtained was dissolved in 100 µL water (4 °C), and the amount of n-Ins-P was determined by its conversion to cyclic PIP. The assay for cyclic PIP synthesis was performed as described [21]. Substrate [^3^H] PGE_1_ was separated from product [^3^H] cyclic PIP through silicic acid column chromatography (50 or 100 µL assay sample was applied on the gel). Elution was performed with 6 mL portions of solvent mixtures (1–6) of increasing polarity. The radioactivity of these fractions was determined; the counts of fractions 4 and 5 were summed up (after quench and yield correction), and the synthesized amount of cyclic PIP was expressed in nmol/assay because of the known specific radioactivity of the substrate PGE_1_. For measuring the activity of cyclic PIP synthase, the incubation time was 10 min; for measuring n-Ins-P, an incubation time of 35 min was sufficient to completely convert all substrate n-Ins-P to cyclic PIP (provided the amount of n-Ins-P was within the assay limits). To calculate the amounts of n-Ins-P synthesized, it is presumed from the chemical structure of cyclic PIP that (a) n-Ins-P and PGE react in a 1:1 ratio to give cyclic PIP, and (b) pr-Ins-P and GTP react in a 1:1 ratio to give n-Ins-P. The assay for cyclic PIP synthesis is performed using [^3^H] PGE_I_ and unlabeled n-Ins-P as substrates (Figure 7A) [21]. In the case of [^3^H]-labeled n-Ins-P, unlabeled PGE_1_ was used (Figure 7B,C).

### 4.6. Fractionation of Rat Liver Homogenate

Fractionation of a rat liver homogenate was performed as described by Wilson and Goulding [36]. The obtained pellets (sedimented at 1100× *g*; 3700× *g*; 16,000× *g*; 107,000× *g*) were suspended in 20 mM Tris/HCl buffer (pH 7.5), containing 1 mM mercaptoethanol and 1 mM EDTA. The samples were frozen in liquid N_2_ and stored at −80 °C. The protein content was determined as described by Smith et al. [37] with a commercially available assay, according to the manufacturer’s instructions.

### 4.7. Synthesis of n-Ins-P on Incubation of Protein-Free Supernatant with Sediment Fraction

Rat liver (11.5 g) was homogenized in 30 mL 10 mM Tris/HCl buffer of pH 7.6 and centrifuged (10,000× *g* for 10 min at 4 °C). The supernatant was separated, and the sediment resuspended in an equal volume of Tris buffer. Supernatant and sediment were divided into two halves, and one part of each was boiled for 10 min at 98 °C, then chilled in ice-water; then, the sediments were separated by centrifugation (10,000× *g* for 10 min at 4 °C). Three milliliter aliquots of the 4 obtained fractions were incubated either separately or in combinations for 15 min at 37 °C. The incubations were stopped by mixing with chloroform/methanol (2:1, *v*/*v*) in twofold excess, and the water phases obtained after centrifugation were lyophilized. The dried samples were dissolved in 0.5 mL of cold water, and the n-Ins-P content was determined in 50 µL aliquots by conversion to cyclic PIP [21].

### 4.8. Degradation of pr-Ins-P by Phosphodiesterase

The assay contained 90 µL pr-Ins-P, 7 µL phosphodiesterase from Crotalus *durissimus terrificus* venom (1 unit/mL) and 103 µL 20 mM Tris/HCl buffer (pH 7.5) in 0.2 mL volume. We withdrew 30 µL aliquots after 0, 2, 5, 10, 15 min, added to 200 µL water and denatured with 400 µL chloroform/ methanol (2:1, *v*/*v*). Remaining pr-Ins-P was determined as described.

### 4.9. Cell Culture of Aortic Myocytes

Aortic cells were cultivated according to [25]. Aortas were removed from anesthetized male rats under sterile conditions and placed in Petri dishes containing Dulbecco’s phosphate buffered saline (PBS; 137 mM NaCl, 2.7 mM KCl, 8.1 mM Na_2_HPO_4_, 1.5 mM KH_2_PO_4_). Aortas were freed of the adventitia and cut into rings of 2 to 3 mm width. Intact aortic rings were explanted in 6-well cell culture dishes (Corning, NY, USA) covered with a thin layer of medium and put into an incubator gassed with 93% O_2_ and 7% CO_2_. The Dulbecco’s Modified Eagle’s medium (DMEM; Gibco, Inchinnan Scotland) was supplemented with 10% (*v*/*v*) fetal calf serum (FCS; Biochrom KG, Berlin, Germany), 2 mM glutamate, 10 mM Hepes, 100 U/mL penicillin, and 100 µg/mL streptomycin. After 3 to 4 h, the rings adhered to the bottom of the dishes and were covered with medium. When cells had grown out from the aortic rings after 10 to 14 days, the rings were removed and cells were transferred (passage 1) into 75 cm^2^ flasks (Nunc, Roskilde, Denmark) using 0.05% trypsin and 0.02% EDTA in PBS. Reaching confluency after 5 to 7 days, the cells were passaged (passage 2) into 24-well dishes. Cells were characterized by immunofluorescence using a monoclonal mouse antibody (Dako, Glostrup, Denmark) against α-actin of smooth muscle cells and by radioligand binding to characteristic receptors (β-adrenoceptors – [^3^H]-(–)-CGP 12,177; K_ATP_ channels – [^3^H]-P1075; and AT_1_-receptors – [^3^H]-Candesartan) [38]. The cells are highly responsive to serotonin and angiotensin II [38]. For labeling of polyphosphoinositides, confluent cell monolayers were washed with FCS-free DMEM and incubated with [^3^H] *myo*-inositol (1 µCi/mL) for 60 h. Thereafter, the cells were stimulated for 30 s with 3 µM angiotensin II, which is known to stimulate the synthesis of cyclic PIP [18]. Then, the cells were briefly rinsed with double distilled water of 4 °C (to reduce buffer salts), scraped off the dishes in 2–3 mL cold water, and were shaken with 4 mL chloroform/methanol (2:1, *v*/*v*). The further work-up is described in preparation and isolation of pr-Ins-P.

### 4.10. HPIC Chromatography of Inositol Bisphosphates

High performance ion chromatography (HPIC) was performed with a DX-500 ion chromatography system (DIONEX, Sunnyvale, CA, USA), consisting of a gradient pump (GP50), an eluent generator (EG40) equipped with an EGC-KOH cartridge, a chromatography oven (LC30) with an internal valve bearing a 100 µL sample loop, and a tunable absorbance detector (AD25). At a constant temperature of 30 °C, an IonPac AG11-HC (4 mm × 50 mm; DIONEX) precolumn and an IonPac AS11-HC (4 mm × 250 mm; DIONEX) analytical column were used. The anion self-regenerating suppressor (ASRS Ultra, 4 mm; DIONEX) was operated in the external-water mode at a current of 300 mA. The Chromeleon 6.2 software (DIONEX) was used for data processing. Deionized water for HPIC was purified using a Millipore system to a specific resistance of 18 MegaOhm or greater. All eluents were sparged with and pressurized under helium. Samples were chromatographed with a mixture of 80% deionized water and 20% (*v*/*v*) methanol as the carrier stream. At a flow rate of 1.0 mL/min, the eluent generator system electrolytically produces the KOH gradient shown in Figure 5A.

## 5. Conclusions

This report shows that the second substrate for cyclic PIP biosynthesis, the activated inositol phosphate, is synthesized from inositol (1:2-cyclic,4)-bisphosphate and preferred GTP. The reaction path, leading from consumed *myo*-inositol to phosphatidylinositol and phosphoinositides, and after hormonal stimulation from inositol (1:2-cyclic,4)-bisphosphate and activated inositol phosphate to cyclic PIP, is the most likely reaction path through which inositol feeding exerts most or all of its positive effects, because cyclic PIP triggers intracellularly many of insulin’s actions [14]. However, since the defect in the signal transduction leading to insulin resistance most likely lies between the insulin receptor and cyclic PIP synthesis [14,19,30], inositol feeding will be helpful as long as no better drugs are found that improve the derailed signal transduction of insulin—or in other words, the hormonal stimulation of cyclic PIP synthesis.

## Figures and Tables

**Figure 1 ijms-25-01362-f001:**
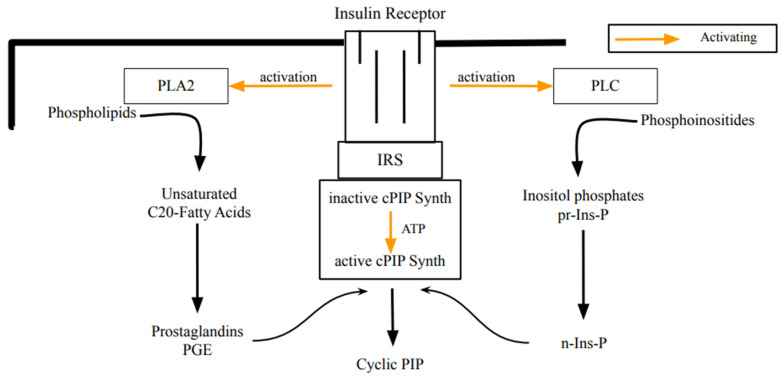
Scheme illustrating the biosynthesis of cyclic PIP by insulin stimulation. Most likely, the insulin receptor tyrosine kinase phosphorylates and activates phospholipase A2, phospholipase C, and cyclic PIP synthase [14]. This initiates the synthesis of the necessary substrates and then the synthesis of cyclic PIP. A comparable scheme, illustrating cyclic PIP synthesis on stimulation with noradrenaline is shown and the hormonal stimulation via G proteins or tyrosine phosphorylation of PLC, PLA2 and cyclic PIP synthase is discussed in [20]. (PLA2, phospholipase A2; PLC, phospholipase C; IR, insulin receptor; IRS, insulin receptor substrate; inactive- or active-cPIP-synth, inactive- or active cyclic PIP synthase; pr-Ins-P, precursor inositol phosphate; n-Ins-P, activated inositol phosphate).

**Figure 2 ijms-25-01362-f002:**
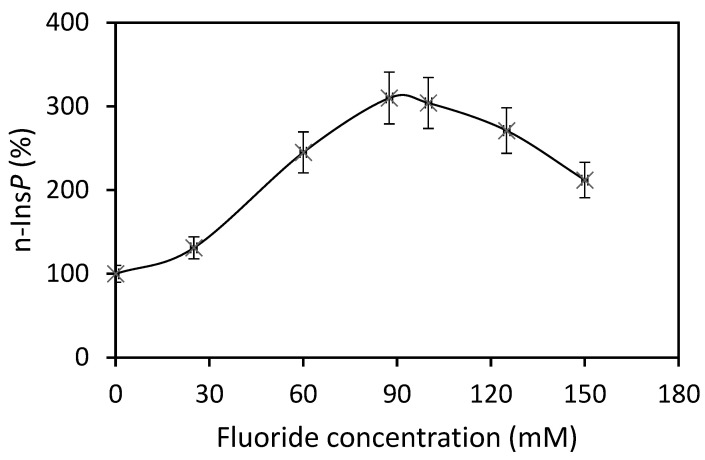
Three-fold increase of n-Ins-P biosynthesis in the presence of fluoride (Results are expressed as means ± standard deviation; n = 3).

**Figure 3 ijms-25-01362-f003:**
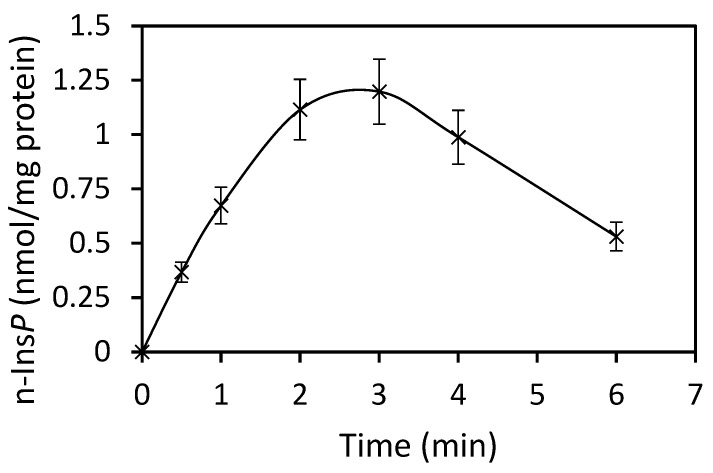
Time course of n-Ins-P synthesis (Results are expressed as means ± standard deviation; n = 6).

**Figure 4 ijms-25-01362-f004:**
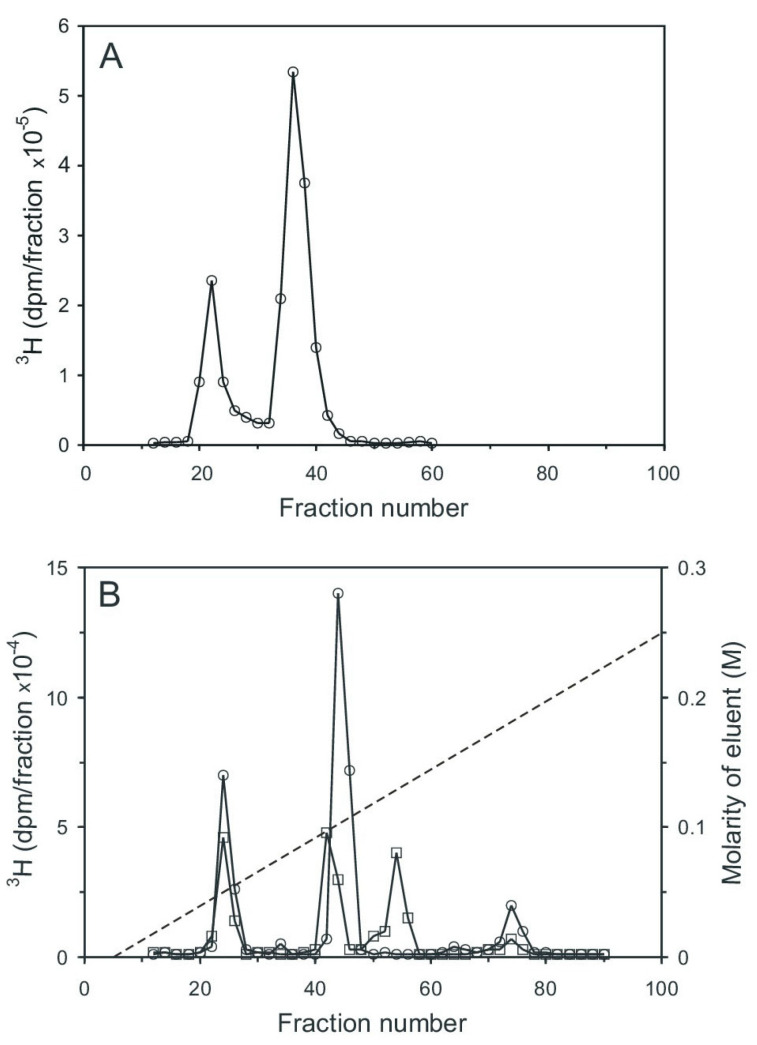
Chromatography of the extract containing [^3^H] inositol-labeled pr-Ins-P. (**A**) Gel filtration on Sephadex G15 (see Section 4.1). 1 mL sample was applied to a 150 mL column (1.5 cm × 85 cm) and eluted with water; 2 mL fractions were collected and their radioactivity content was determined. (The void volume of the column was reached in fraction 18 and the column volume in fraction 75). (**B**) Anion-exchange chromatography on a HL-Q-Sepharose HP column (1.5 cm × 25 cm). Material of the fractions (20–28) of the gel filtration (**A**), obtained from unstimulated control cells (squares) and from cells stimulated for 30 s with 3 µM angiotensin II (circles) was chromatographed. The dashed line indicates the linear salt gradient. (The results shown are representative of three independent experiments.).

**Figure 5 ijms-25-01362-f005:**
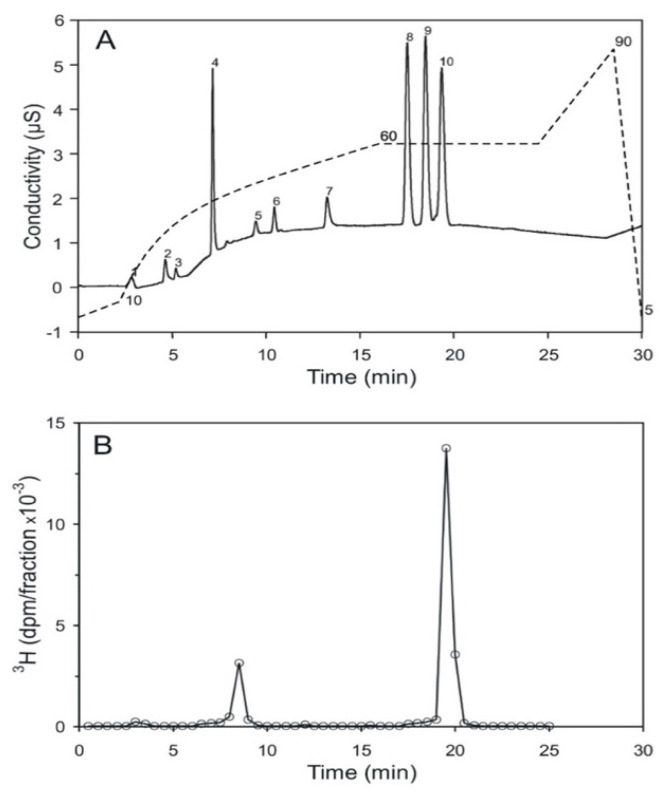
Separation of inositol bisphosphates by HPIC (high performance ion chromatography). (**A**) The positional-isomers, inositol (1,6)-, inositol (4,5)-, and inositol (1,4)-bisphosphates, corresponding to the peaks 8, 9, and 10, respectively, were detected by conductivity measurement; peak 4 is attributed to chloride ions; the dashed line indicates the KOH-gradient. The detection limit is 10 pmol and 100 pmol per substance were applied to the column. (**B**) Elution profile of [^3^H] inositol-labeled, acid-hydrolyzed pr-Ins-P. 86% of the radioactive material applied to the column eluted as inositol-(1,4)-bisphosphate; the small peak eluting after 8.5 min eluted in the range of inositol monophosphate. The pmol amount of the applied, hydrolyzed inositol phosphate was not determined (The result shown is representative of two independent experiments.).

**Figure 6 ijms-25-01362-f006:**
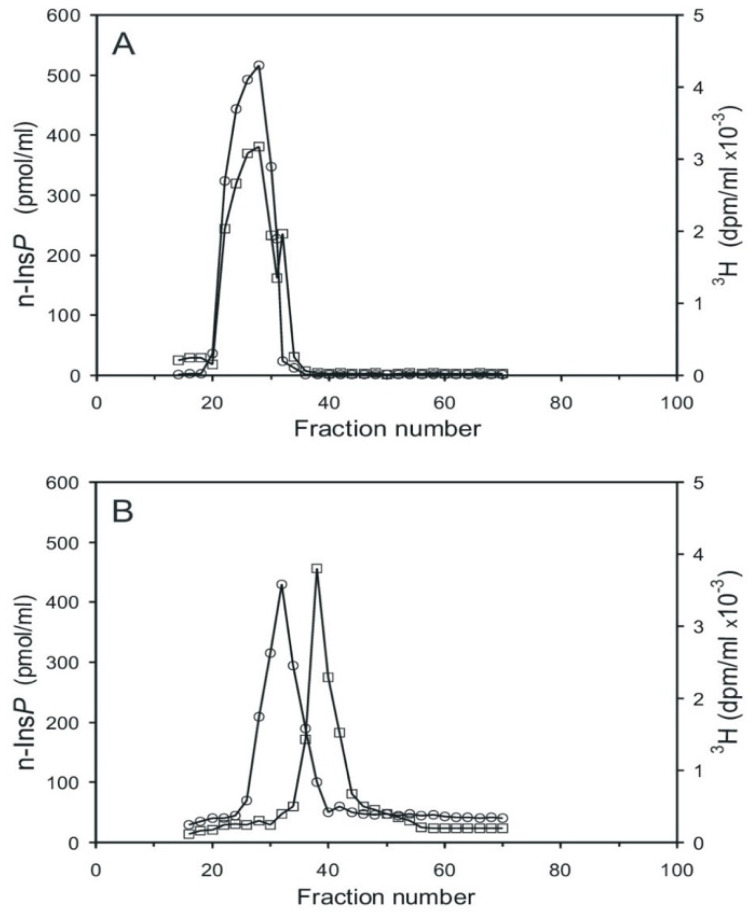
Chromatography of [^3^H] guanosine-labeled n-Ins-P mixed with unlabeled n-Ins-P on Sephadex G15. (**A**) Re-chromatography of [^3^H] guanosine-labeled n-Ins-P. [^3^H]-radioactivity (curve with squares) and n-Ins-P activity (curve with circles). (**B**) As control [^3^H] GTP was incubated without pr-Ins-P and no [^3^H] guanosine-labeled n-Ins-P was obtained (The results shown are representative of two independent experiments.).

**Figure 7 ijms-25-01362-f007:**
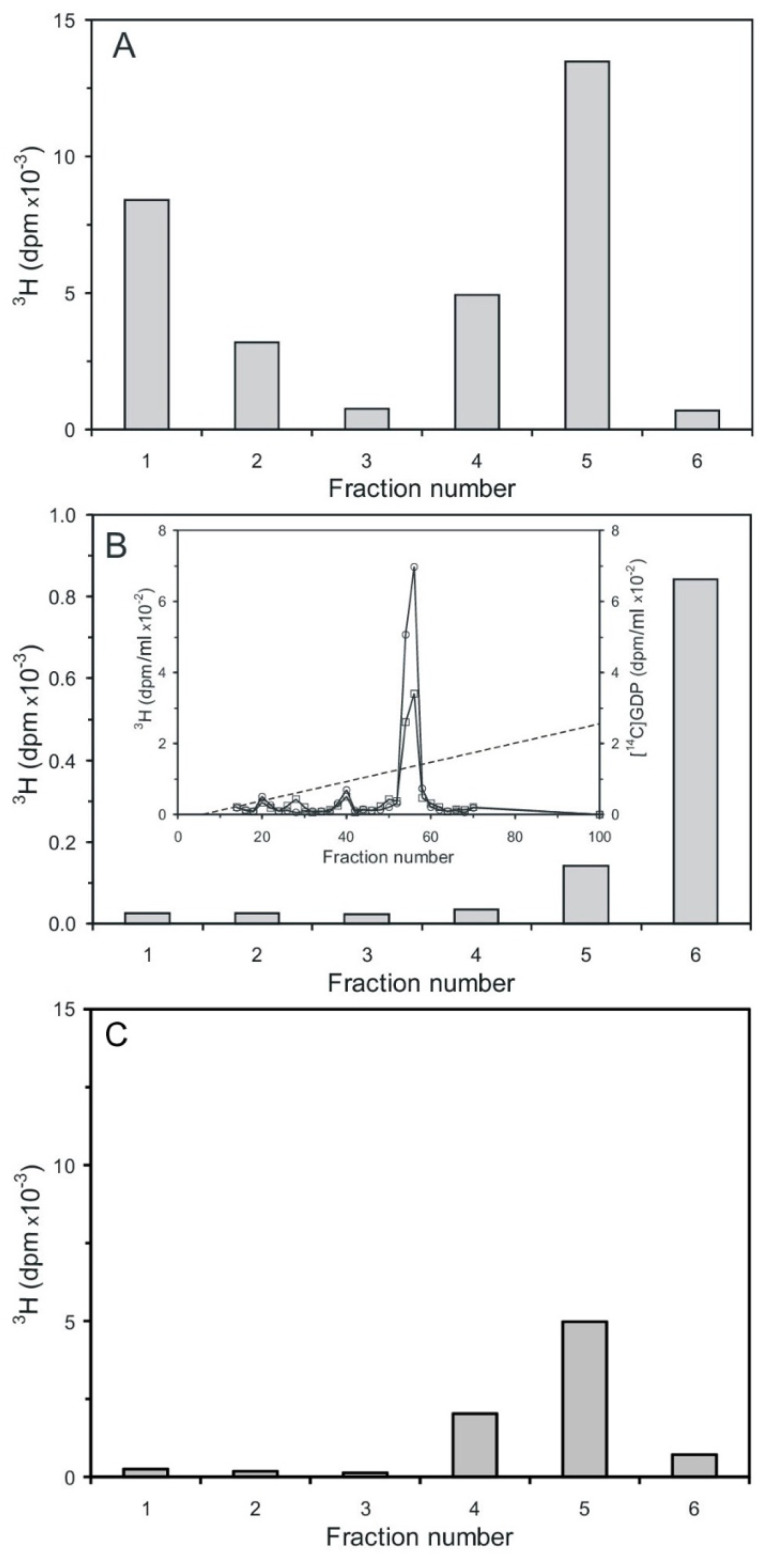
Separation of substrates and product cyclic PIP by silicic acid chromatography [20,21]. (**A**) Separation of [^3^H] cyclic PIP from [^3^H] PGE_1_. Residual PGE_1_ elutes in fraction one and two and synthesized cyclic PIP elutes in fractions four and five. (**B**) Separation of cyclic PIP using unlabeled PGE_1_ and [^3^H] guanosine-labeled n-Ins-P in the assay. No [^3^H] guanosine-labeled material eluted in the cyclic PIP containing fractions 4 and 5. All [^3^H] guanosine-labeled material eluted with water in fraction six. The inset shows the anion-exchange chromatography of this product (squares), which co-chromatographs with [^14^C] GDP (circles). The dotted line indicates the linear salt gradient. (**C**) Separation of cyclic PIP using unlabeled PGE_1_ and [^3^H] inositol-labeled n-Ins-P. All [^3^H] inositol-labeled product elutes in fractions 4 and 5 in which cyclic PIP elutes. In the assay of (**A**) using the specific radioactivity of the substrate PGE (1.7 nmol/30,000 dpm), the amount of synthesized cyclic PIP can be calculated (1.02 nmol). In the assay of (**B**) from the specific radioactivity of obtained [^3^H] n-Ins-P (6.2 dpm/pmol) can be calculated that 162 pmol [^3^H] GDP were obtained. That means that 162 pmol cyclic PIP were obtained, assuming that the equation of cyclic PIP synthesis is correct. In the assay of (**C**), the tritium-labeled myo-inositol is diluted by an undetermined amount of the cells’ own inositol synthesis; that means in this case, the synthesized amount of cyclic PIP cannot be expressed in pmol (The results shown are representative of three independent experiments).

**Figure 8 ijms-25-01362-f008:**
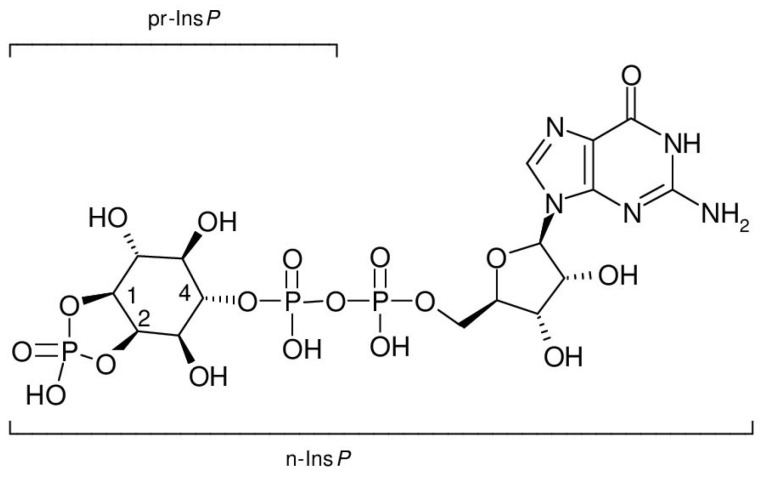
Proposed chemical structure of n-Ins-P, guanosine diphospho-4-inositol (1:2-cyclic)-phosphate (lower bracket). The upper bracket indicates the incorporated precursor inositol phosphate, inositol (1:2-cyclic,4)-bisphosphate.

**Table 1 ijms-25-01362-t001:** Dependence of inactivation of n-Ins-P on the ionic strength of the solution.

Salt Concentration (M)	Decay (%)
0.0	0
0.1	20
0.25	57
0.5	77
1.0	93

A constant amount of n-Ins-P was incubated in 10 mM Tris-HCl buffer of pH 7.5, containing an increasing amount of sodium chloride; after 30 min incubation the remaining n-Ins-P was determined.

## Data Availability

We tried to report all experimental details in the Section 4. In case further questions arise, please inform the corresponding author, he will be engaged to answer questions or forward any questions to the coauthors.

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
