# Peer review of "Activated Inositol Phosphate, Substrate for Synthesis of Prostaglandylinositol Cyclic Phosphate (Cyclic PIP)—The Key for the Effectiveness of Inositol-Feeding"

_ijms, 2024, doi:10.3390/ijms25031362_

Round 1

Reviewer 1 Report

Comments and Suggestions for Authors

Please find below my general impressions and specific comments about the manuscript.

In the manuscript, the authors conducted numerous experiments to investigate the biosynthesis of activated inositol phosphate. The manuscript is well-structured and presents a substantial amount of data with a commendable level of relevance. My recommendation is for its acceptance with minor changes.

(1) The title could be more concise and explanatory.

(2) The graphs lack error bars. The authors should include the standard deviation for each measurement in every graph, mainly in Figures 1 and 2.

(3) The software used for plotting the graphs and conducting statistical analysis was not disclosed.

(4) It is suggested that the authors include a figure or scheme illustrating the metabolic pathway under investigation in the introduction. This addition will enhance the manuscript's clarity and attractiveness for readers.

Author Response

Dear Referee 1,

For reviewing our manuscript, I thank you very much. I also thank you for the suggestions to improve the manuscript, to which I respond as follows:

The title is shortened and reads now: Activated inositol phosphate, substrate for synthesis of prostaglandylinositol cyclic phosphate (cyclic PIP). The key for the effectiveness of inositol-feeding.

Error bars in the previous Figures 1 and 2, now Figures 2 and 3 are added.

The used software for plotting was Microsoft Excel. This is mentioned in the material section 4.1

In the introduction, a new Figure 1 is added, which shows how cyclic PIP and its substrates PGE and n-Ins-P are synthesized on stimulation with insulin.

I hope that the made corrections meet your expectations. Once more thank you for your help, kind regards,

Heinrich Wasner

Reviewer 2 Report

Comments and Suggestions for Authors
In this manuscript, Antonios Gypakis describes the biosynthesis of activated inositol phosphate from inositol (1:2-cyclic,4)-bisphosphate. The manuscript is well-written; however, some issues need to be addressed:
1. The manuscript's main goals are unclear. Please include a detailed sentence outlining these goals.
2. To better understand the synthesis of n-Ins-P, I suggest adding a pathway figure for the readers.

3. Line 241: There is a different font size.

4. Line 346: The reaction sequence is not clear. Please put this reaction in a separate figure. 

5. Figure 7: The structure does not lack symmetry (The inositol moiety does not appear to be a symmetric cyclohexane.); please fix it. In addition, the phosphate groups should have a negative charge.

Author Response

Dear Referee 2,

For reviewing our manuscript and also for your the suggestions how to improve it, I thank you very much. According to your suggestions the following corrections were made:

The main topic of the report is to characterize the biosynthesis of activated inositol phosphate. This is mentioned at the end of the introduction. Presently, it is a hot topic to discover curing effects of inositol supplementation. Cyclic PIP and the synthesis of the substrate activated inositol phosphate are the most likely way which can explain this effect. This goal is mentioned at the end of the introduction.

In the discussion a new Figure 1 is added, which shows how the substrates for the synthesis of cyclic PIP and also the synthesis of cyclic PIP is stimulated by insulin.

I am very sorry that on the preliminary editing of the manuscript, because of my imperfect typing, this derailment happened. I corrected the font size. The reaction scheme, which shows how cyclic PIP is synthesized, is now in a stable version added.

I am very sorry that I did not succeed to improve the bend six ring of the inositol in the Figure 8. The chemical drawing program, which we have, bends this six-ring, when we close the 5-ring phosphodiester. I ask that this can be tolerated.

I thank you once more for your help to review this manuscript and also for suggesting corrections in order to improve it.

Kind regards,

Heinrich Wasner

Reviewer 3 Report

Comments and Suggestions for Authors

The presented study is an interdisciplinary study detailing the assay for biogenic phosphates. There are a few points that the author needs to address.

1. The chromatography detailing could have been improved so that one can reproduce the method.

2. Were standards used for chromatography?

3. Loading concentration: Was there any specific limit to loading a particular amount for HPIC?

4. Author stated that "Separation of inositol bisphosphates by high-performance ion chromatography, where peaks 8, 9 and 10 corresponds to the positional isomers, [inositol (1,6)-bisphosphate, inositol (4,5)-bisphosphate and inositol (1,4)-bisphosphates respectively]. How did the author find out these peaks specific to this bisphosphate? Was there any method that was previously discussed? Please specify or state clearly.

5. The paper is not organized in its current form, as it is hard to follow up. Please organize them; the material and method should not be detailed in the end. Please check the journal guidelines.

Author Response

Dear Referee 3,

For reviewing our manuscript and also for your suggestions how to improve it, I thank you very much. According to your suggestions I made the following changes of the manuscript:

In the methods section in point 4.4 details of the applied chromatography are given and also informed in which fractions used standards elute.

In the legend of Figure 5 is the loading concentration given, which was applied for separation of the three inositol phosphates by high performance ion chromatography. In the text in lines 226 - 227 is informed how the three peaks were assigned to the different inositol bisphosphates.

I am sorry that I had put the conclusion before the Methods Section. It is now after this section at the end of the text file.

I hope that the made corrections meet your expectations. I thank you once more for helping to evaluate and improve this manuscript.

Kind regards,

Heinrich Wasner

Reviewer 4 Report

Comments and Suggestions for Authors

In the manuscript, the authors report that cyclic PIP may be a key factor for explaining the curative effect of inositol feeding that helps patients with metabolic conditions related to insulin resistance. The study is interesting and may provide useful information for improving the derailed signal transduction of insulin. The manuscript is recommended for publication after some issues being addressed by the authors.

1. The full name for some short forms like the enzyme, PLC, should be given.

2. Some typos found such as line 77, Last not least.

3. The “,” and “.” for decimal point are not consistently used. This makes confusing for the numbers, such as line 128: (1,100 g, 3,600 g, 16,000 g, 107,000 g). It is not clear whether it is 1100 g or1.100 g.

4. It is unclear how the synthesized [3H] guanosine-labeled n-Ins-P is characterized and quantified.

5. Fig. 3B, the authors showed that angiotensin II stimulated cyclic PIP synthesis rat’s cells. And, the authors further extend to a conclusion that hormonal stimulation increases cyclic PIP synthesis. Could this statement be further verified with human cells?

Comments on the Quality of English Language

Some sentences are too long and complicated and that makes readers not readily to pick up the meaning. It could be much better if the authors make it more readable.

Author Response

Dear Referee 4,

For reviewing our manuscript, I thank you very much. I also thank you for the suggestions how to improve the manuscript. I made the following changes (all made changes are marked yellow):

The full name and the abbreviation in brackets are given. Only for the nucleotides I did not do this and hope that this is okay.

I am sorry that I overlooked this, it is corrected now and reads: last but not least.

I deleted the colons in the used numbers in order not to confuse any reader.

In the legend to Figure 7 is mentioned how the amount of [3H] n-Ins-P is determined and how much cyclic PIP is synthesized with the substrate.

The hormonal stimulation of cyclic PIP synthesis: Two sentences are added in the introduction. Now is also mentioned briefly the structure of cyclic PIP in the introduction.

I have tried to shorten the sentences and hope that I found at least most of them. I am sorry that Germans like to make long sentences.

Once more thank you for helping to evaluate and improve this manuscript.

Kind regards,

Heinrich Wasner